# Influence of the Slaughter Method on the Hygienic Quality of Beef Cattle Meat and Animal Welfare Biomarkers

**DOI:** 10.3390/ani13061014

**Published:** 2023-03-10

**Authors:** Said Bouzraa, Estrella I. Agüera, Francisco Requena, Inmaculada Rodríguez, Salud Serrano

**Affiliations:** 1Food Science and Technology Department, Faculty of Veterinary, University of Córdoba, 14071 Córdoba, Spain; 2EGCH_Halal Institute Spain S.L., 14002 Córdoba, Spain; 3Cellular Biology, Physiology, and Immunology Department, Faculty of Veterinary, University of Córdoba, 14071 Córdoba, Spain

**Keywords:** stunning, bovine meat, halal, microbiology, CK, LDH, glucose, cortisol

## Abstract

**Simple Summary:**

The objective of this study was to investigate the influence of the slaughter method on the microbiology of the meat and the animal welfare physiological indicators for beef cattle. Three slaughter procedures were compared, as follows: regular slaughter (with a penetrative captive bolt), halal slaughter, and halal slaughter with a non-penetrative captive bolt (reversible stunning accepted by some halal religious authorities). We conclude that the halal slaughter with stunning showed the best results for microbial counts (enterobacteria and coliforms) and for the considered biomarkers of animal welfare, lactate dehydrogenase, glucose, and creatine kinase; however, this slaughter type gave the highest amount of cortisol.

**Abstract:**

The main objective of this research was to carry out a comparative study between different types of slaughter in beef cattle with and without stunning. In addition, the hygienic quality of the obtained meat was determined through microbiological analysis and the animal welfare at the time of slaughtering was assessed by means of physiological parameters. A total of 52 blood samples collected at the slaughterhouse during slaughter (10 for each type of slaughter: regular, halal, and halal with stunning; 10 at the time of resting; and 12 from rotating box slaughter) were analysed for physiological parameters indicating animal welfare status, namely, glucose, cortisol, lactate dehydrogenase (LDH) and creatine kinase (CK). In addition, the meat from 30 of the above animals was analysed for aerobic mesophilic bacteria, enterobacteria and coliforms. Moreover, a radiological study of the possible skull damage due to the non-penetrative captive bolt used at the time of stunning in the halal rite slaughter was carried out. A significance difference (*p* < 0.05) in the microbiological counts per type of slaughter was observed. It was proven that the amounts of glucose, LDH, CK, and cortisol in plasma were influenced (*p* < 0.05) by the type of slaughter. The halal rite slaughter using stunning with a non-penetrative captive bolt resulted in the best hygienic quality of meat and obtained the lowest values for all animal welfare biomarkers.

## 1. Introduction

The “animal welfare” (AW) concept appeared as a concern for farming animals in European countries in the mid-20th century. It arose as a response to intensive breeding systems. It is in the last 25–35 years that interest in AW has been promoted. Animal welfare is an ambiguous concept, difficult to define from a scientific point of view, as it derives from an ethical concern of social origin. A definition would be “the state of the animal being in harmony with the environment, having physical and mental health, meeting its specific needs [1].

Although current legislation (Regulation (EC) 1099/2009) [2] authorizes religious slaughter, exceptionally without stunning, it remains a controversial issue from the point of view of AW with enormous social implications. Halal meat is defined as meat obtained from slaughtered halal animals and processed in accordance with Islamic dietary laws (Hadith No. 668 of Sahih Bukhari, Vol. 3, Book 44). As [3] exposed, seven principles guide a halal slaughter: (1) the animal must be of a permitted species; (2) the slaughtering process must be conducted by a practicing Muslim who is of sound mind; (3) the person who is performing the slaughter must make the intention of performing the slaughter then recite a blessing, which is typically “Bismillah and Allahu Akbar” or “In the name of Allah and Allah is Greatest”; (4) animals must be alive before sticking (killing); (5) use a sharp knife; (6) stick the front part of the neck, severing the carotids, jugulars, trachea, and oesophagus without reaching the spinal cord; and (7) the blood must be drained to achieve a complete bleeding (Quran, 6:118–119; 16:115; Hadith No.17 of Imam Nawawi by Sahih Muslim).

From the scientific viewpoint, there exists certain controversy on the slaughter type from the point of view of the AW, according to the consulted sources [4]. Thus, [5] defended halal slaughter as a compatible method with animal welfare in their comparative study of different slaughter methods (without and with stunning). In the same way, [6] concluded that the slaughter without stunning is the most natural and least traumatizing method. Other study [7] noted that opponents of pre-slaughter stunning for halal slaughter have often cited the possibility of animals dying following stunning and before exsanguination as the main reason pre-slaughter stunning contradicts the Islamic dietary rules. Other authors [8] are of the opinion that the stunning of animals prior to slaughter results in the retention of more blood in the carcass in comparison with those slaughtered without stunning. In this sense, [9] set the amount of blood left within the carcass after bleeding is one of the most significant factors affecting the level of contamination and thus increases the degree of the deterioration. However, several studies have demonstrated that there is no difference between animals that are slaughtered either with or without pre-slaughter stunning in terms of the total blood lost at exsanguination [10,11,12].

Nowadays, the quality of food is determined not only by the overall nature and safety of the final product, but also by the welfare status received by the animal producing the food [13]. The fact that improving AW can positively affect the quality of products of animal origin, and disease prevention, also has a direct influence on food quality and food safety [14]. The AW in animals for production can be predicted by the alteration of endocrine markers like cortisol [15,16] and other biochemicals (glucose and lactate dehydrogenase (LDH)) [17,18,19,20]. However, physiological indicators have not been investigated enough at the time of slaughtering because samples require additional handling, and testing is expensive [21]. For this reason, in this research we considered that it would be interesting to evaluate these parameters.

As the stunning or not stunning system at the time of slaughtering is crucial for the maintenance of the AW, the aim of this study was to evaluate the influence of the slaughter method on the hygienic quality of beef cattle meat and on the animal welfare by measuring biomarkers.

## 2. Materials and Methods

### 2.1. Sampling and Slaughtering Conditions

A total number of 52 animals from two different slaughterhouses were studied. Thirty animals from the Cooperativa Ganadera del Valle de los Pedroches (COVAP) sited in Pozoblanco (Córdoba, Spain) provided blood and meat samples (diaphragm pillars): one group of animals (*n* = 10) was slaughtered with a penetrating captive bolt stunning (regular stunning process, RS), another group (*n* = 10) was slaughtered without stunning (halal slaughtering, HS), and a last group of animals (*n* = 10) was slaughtered by a halal authorized stunning (accepted by some halal religious authorities) with a non-penetrative captive bolt (HSS). All these animals were slaughtered using upright restraint. The slaughtered animals were males Limousine beef breed aged 12–14 months (weight 610–650 kg) and came from the same farm, with similar handling conditions. Ten more animals with the same origin (similar breed, gender, age, and weight) provided blood samples at the time of resting at the lairage in the same slaughterhouse. The second slaughterhouse (Frimancha, Ciudad Real, Spain) was used to obtain 12 more blood samples from beef cattle (same gender, breed, age, and weight) slaughtered by halal rite with a rotating restraint box system (RRB).

The penetrative captive bolt stunning (RS or regular stunning process) renders the animal immediately deeply unconscious for a prolonged period. This method penetrates the encephalon producing an irreversible damage in the animal. It has been unanimously found to be a method of stunning for cattle in accordance with animal welfare provisions according with European regulations [2]. The thickness of the cartridge used must be aligned with the size of the animal to ensure that the full length of the bolt penetrates the animal’s skull. However, there is a different kind of stunning, reversible non-penetrative captive bolt, for halal slaughtering, which, according to some halal standards, can be acceptable with a strict control and checking for each skull to evaluate the damage caused, and some carcasses can be rejected.

A Karl Schermer Type KC (Germany) non-penetrative captive bolt was used for reversible stunning for halal slaughtering (HSS): following ignition of the cartridge, the propellant charge accelerates the bolt to such a strong extent that the impact plate strikes the skullcap of the animal at a speed of about 45–65 m/s.

The captive bolt stunners were used solely with the original cartridges (type: Calibre 6.8/15) of Karl Schermer GmbH & Co., KG.

During exsanguinations, disposable vacutainer tubes with anticoagulant (ethylenediaminetetraacetic acid) were used to collect 25 mL of blood samples from the jugular vein immediately after throat cutting for HS and HSS. In animals slaughtered with RS system the blood was collected at the chest entrance cut. The animals were then bled for between 6 and 8 min while they were still hanging. Each blood sample was kept in ice until analysed (maximum 6 h after sampling).

### 2.2. Microbiological Analysis

Ten grams of each meat sample were homogenized into 90 mL of peptone water solvent and decimal dilutions were prepared using the same solvent. Aerobic mesophilic bacteria (ISO 4833-2) were counted onto a standard plate count agar (PCA) and incubated at 30 °C for 72 h. Total enterobacteria (ISO 21528-2) was counted onto violet, red bile glucose agar (VRBG) incubated at 37 °C for 24 h. Finally, total coliforms (ISO 4831) were determined in brilliant green lactose bile broth, incubated at 31 °C for 24 and 48 h. Determinations for all microbiological analysis were carried out in duplicate.

### 2.3. Physiological Parameters Determinations

The 52 blood samples were centrifugated (1000× *g* for 12 min at 17–24 °C) and plasma decanted and stored at −20 °C. The samples of stored plasma were analysed for creatine kinase (CK), glucose, cortisol, and lactate dehydrogenase (LDH) using an immunoassay analyser (Model Cobas 6000-C501; Roche, Japan) with commercial kits for CK (kit ref 07190794190, Roche), glucose (kit ref 04404483190), cortisol (kit ref 06687733190), and LDH (kit ref 03004732122). The quantitative determination of CK and LDH activities was expressed in units per litre (U/L) in plasma. Determinations of glucose and cortisol were expressed in milligrams per decilitre (mg/dL), and micrograms per decilitre (mcg/dL), respectively.

### 2.4. Methodology of Radiographic Study

Digital radiographs of the lateral head were obtained on beef cattle stunned by the non-penetrative captive bolt (10 HSS samples), to determine the skull damage of the forehead of the animal, according to Appendix A2 of Malaysian Protocol [22] for the Halal Meat and Poultry Productions (MS-1500:2009). A standard approach for radiographic examination (PotroDR1^®^, CVM, Metron Software) was used. For each region, a minimum of 3 standard views were obtained, with 72 MHz and 20 MAs. The macroscopic damage in the frontal bone was observed.

### 2.5. Statistical Analysis

Statistical analysis tested whether the three different slaughtering types were associated with microbial counts, and cortisol, glucose, LDH, and CK contents. All data were collected from two replicates.

The data for microbial counts, and plasma levels of cortisol, glucose, LDH and CK did not agree with parametric assumptions because they were non-normally distributed. For this reason and as more than two groups are being compared, the Kruskal–Wallis test was used to identify the statistically significant differences between the slaughtering systems.

The data analysis was performed using the software IBM SPSS Statistics for Windows version 25 (IBM Corp., Armonk, NY, United States. 2017) and setting confidence intervals of 95% (*p* < 0.05). After the Kruskal–Wallis test, the post-hoc Mann–Whitney U test was carried out for all comparisons to see how the groups differed.

## 3. Results

The results from the Kruskal–Wallis test revealed significant statistical differences between slaughtering systems and the parameters studied. The slaughtering system influenced the plasma levels of glucose (*p* < 0.000), cortisol (*p* < 0.000), LDH (*p* < 0.000), and CK (*p* < 0.001); it also influenced enterobacteria (*p* < 0.000), aerobic mesophilic bacteria (*p* < 0.004), but not coliforms (*p* > 0.61).

With respect to the microbiological parameters, the HSS slaughtering type showed the lowest values for enterobacteria and coliforms in comparison to HS and RS (Figure 1). The results obtained showed that there were significant differences between the median values of HSS and HS (*p* < 0.003) and RS (*p* < 0.000) types for total enterobacteria and aerobic mesophilic bacteria (*p* < 0.004).

In relation to physiological welfare indicators (Figure 2), HSS slaughter showed significant differences (*p* < 0.005) for all parameters. Regarding the LDH plasma levels, there were differences between HSS and resting (*p* < 0.001) and RRB (*p* < 0.000) and for plasma CK levels the differences were when comparing HSS to resting (*p* < 0.000). The HSS slaughtering system was observed to be the one that presented the lowest plasma levels for CK, LDH and glucose; however, this type of slaughter showed the highest plasma level of cortisol and, consequently, the statistical analysis revealed significant differences between HSS and resting (*p* < 0.000), HS (*p* < 0.011) and RS (*p* < 0.018). Our plasma glucose results showed that the increase of its levels in plasma was an indicator of maximum stress, with the RRB the most stressful scenario, and statistically significant differences were found between this slaughtering system and HSS (*p* < 0.000), resting (*p* < 0.000) and HS (*p* < 0.045).

Descriptive statistic can be observed in Table 1 and Table 2.

Regarding the radiographical study, it was observed that the skulls presented rating 1 “no visible damage” (70%) or 2 “indentation no cracking” (20%), and rating 3 “indentation with cracking but no displacement” (10%) following the Malaysia Standard (Figure 3, Figure 4, Figure 5). Consequently, 90% of the HSS animals (ratings 1 and 2) would be accepted for halal commercialization. However, no radiographical differences existed between ratings 1 and 2 (Figure 3 and Figure 4); only macroscopic differences in soft tissues were noted.

## 4. Discussion

Many researchers have reported a correlation between meat quality and blood. The more blood retained, the poorer the meat quality [23,24]. Moreover, De Oliveira [25] showed that bleeding should be efficient to guarantee meat quality. Therefore, it is essential to reduce the risk of carcass contamination with blood, which serves as a perfect medium for bacteria growth [26]. Meat with less blood was found to be lacking some nutrients [8]; this extends shelf-life of the meat thereby reducing product deterioration. Moreover, excessive stress to the body seemed to increase blood splash and reduce bleeding [27]. Additionally, Hayes et al. [28] showed that high levels of handling stress may increase the time required for the animal to become unconscious and may possibly have negative effects on post-mortem muscle metabolism.

There is no consensus about the influence of stunning on the bleeding phase. Some authors [9] found that a neck incision for halal slaughter without stunning for rabbits resulted in higher blood loss compared to gas stunned rabbits. On the contrary, Sabow et al. [29] affirmed that slaughtering goats following minimal anaesthesia did not result in poor bleed-out compared to slaughtering fully conscious goats and did not affect the keeping quality of meat. Our finding in which the HSS slaughtering system was observed to be the one that presented the lowest microbial counts, showed that stunning was the higher blood loss method. Several researchers could not establish any differences in rate of blood loss amongst animals slaughtered by incision (traditional halal method), and those stunned prior to incision [9,11,30].

Some studies on chickens [8] also reported that higher blood loss in halal slaughter was associated with lower bacteria count in minced meat at 48 h post-mortem. The Islamic hanging slaughtering method for chicken resulted in higher level of bleeding compared to Islamic traditional slaughtering and electrical stunning methods. Furthermore, [30] indicated that glucose in the blood serves as substrate favourable for microbial growth such as *Pseudomonas*, which grows in meat easily in the presence of blood. Moreover, experiments by [9] suggested that residual blood was found to be less in the carcasses of rabbits slaughtered using a halal method that in turn produced lower bacteria counts in the longissimus lumborum.

Some authors [31] assert that the compliance of any method of stunning depends on whether the animal remains alive (is able to recover and live a normal animal life if not slaughtered) following the stunning (Hayat Mustaqirrat) and prior to slaughter or not, whether the act of stunning in itself is painful or not to the animal being stunned, and whether or not the stunning affects the flow of blood after slaughter—if it meets all these requirements, then it is permissible. If it does not, then the process would be considered Makrooh/undesirable [32]. In this sense, [33] indicated that restraining the animal in a comfortable upright position using a modified American Society for the Prevention of Cruelty to Animals (ASPCA) pen before and during slaughter was less stressful than shackling, inverting, or hoisting. Our results confirmed a previous study by [34] showing stress responses such as cortisol levels and haematocrit values of cattle subjected to religious slaughter with the Weinberg pen, in which the animal is inverted, were significantly higher than those of cattle slaughtered in the ASPCA pen, in which the animal is standing. Additionally, the average time spent in the Weinberg pen was eight times longer than the time spent in the ASPCA pen [35]. Our results consider the halal slaughter with rotating restraint box (RRB) as the worst procedure in view of the biomarkers of AW. Similar consideration of this method of restraint was found by [36] with the highest levels of both struggling and vocalization that were observed when cattle were turned on their sides, compared to when they were restrained in the upright position.

With respect to the HSS method, even if a very light stun that does not damage the skull is used, in our study the cut is carried out immediately after stunning, and the use of the upright box allows it without any delay. Therefore, the time to bleed is not delayed avoiding the possibility of the animal returning to consciousness.

As Majeed et al. [37] exposed, a beating heart is indispensable for a thorough bleed-out by the animal and attainment of a higher amount of blood at exsanguination is better during halal slaughter according to Shari’ah (halal law). Even if the variation in Islamic jurisprudence is one of the primary determinants of intra-regional trade of halal meat import demand in OIC member countries [38], most certifiers indicate that they accept pre-slaughter stunning if the stunning does not result in the death of animal prior to exsanguination, a Muslim should perform the slaughter and a short prayer must be recited and only manual (by hand) slaughter is acceptable [39]. According to this premise, our findings revealed the halal slaughter with stunning was the best procedure from the microbiological and AW point of view. At present, the Malaysian Protocol for the Halal Meat and Poultry Productions allows the non-penetrative captive bolt stunning for bovine [22] setting a rating scale of 6 grades depending on the skull damage. The radiological study in our research demonstrates the benefits of this stunning method in the tested animals, evidencing the ratings 1 or 2 for 90% of the skulls; this means the halal condition for the carcass.

With respect to the physiological biomarkers of the AW, [16] in their study of cattle temperament and handling conditions stated that biochemical changes supported the idea that special care should be taken for managing. Plasma glucose and protein concentration were associated to management conditions, suggesting the possibility of a favourable effect of resting time before slaughter. The increment of plasma cortisol levels at slaughter, independently of management or temperament characteristics, suggested an important effect of stress associated to slaughter procedures, a clearly key issue to improve. However, in our study, this AW indicator would reveal minimum stress or suffering with high plasma level increase.

Our results showed a high plasma level of CK (mean 3256.4 U/L) and LDH (mean 1701.4 U/L) at the resting time, which might be due to stressful handling conditions. On the contrary, both halal slaughters (HS and HSS) resulted in the lowest plasma levels of CK (142.1 and 378.1 U/L, respectively) and LDH (1399.9 and 1027.9 U/L, respectively). Some authors [34,40,41] have exposed that the increased levels of CK in the plasma are an indication of how stressful the handling facilities were before the animal was slaughtered and the extent of muscular damage during handling. Moreover, the presence of this enzyme in the plasma is due to breed temperament, excitability and fighting. This enzyme is mostly located in different tissues and its presence in the plasma serves as an indication of muscle damage [42]. Stressors due to physical exertion are normally measured using CK and LDH as these are found in the muscles (sarcomere length) and muscle damage consequently affects meat colour and tenderness [40,43,44,45,46]. However, [47] pointed out that CK and LDH can be used as indicators of welfare in slaughter cattle but cannot be used to predict the quality of meat.

With respect to glucose, when animals are exposed to stressful conditions, they secrete catecholamine and glucocorticoids, which enhance hepatic glycogenolysis, thereby leading to high glucose levels [48]. Our results coincide with those of [6] that measured the plasma level of glucose in the moments prior to the death of the calves, obtaining 0.7 g/L in halal slaughtering and 1.5 g/L in regular stunning and suspended bleeding animals. Our study reveals the lowest plasma level of glucose (mean = 53.4 mg/dL) for the HSS slaughter.

## 5. Conclusions

The importance of this research is based on the necessity of obtaining data about AW conditions at the time of slaughtering under the point of view of religious slaughter features, especially when the European regulation allows the non-stunning procedure, which might be against the AW. Our results reveal the use of the non-penetrative captive bolt as the optimal method of stunning in the halal slaughtering in beef cattle from the point of view of the animal welfare and the microbiological quality of the meat. Widespread authorization of this type of stunning that keeps the animal alive (although stunned) until the moment of death by slitting its throat would allow halal slaughter of cattle in countries where it is currently not authorized due to the fact that no stunning is introduced. Certifying entities, meat companies and, in general, the halal market need to resolve the controversial issue of animal slaughter within an environment that prioritizes animal welfare. Further research is warranted in order to evaluate acute handling stress and handling protocols in order to improve welfare perspective in the beef production systems.

## Figures and Tables

**Figure 1 animals-13-01014-f001:**
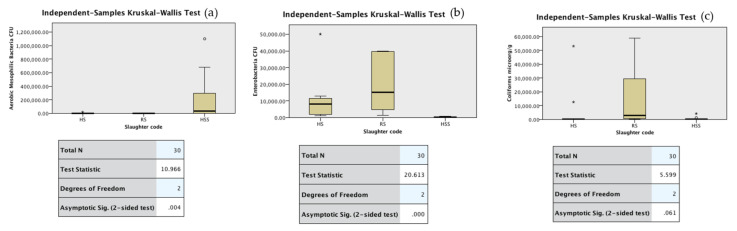
Box and whisker plots for aerobic mesophilic bacteria (**a**), total enterobacteria (**b**), and total coliforms counts (**c**) comparing halal slaughter (HS), halal stunning slaughter (HSS), and regular slaughter (RS). In the boxplot, the thicker line in the middle is the median value. The top and bottom box lines show the first and third quartiles. The whiskers show the maximum and minimum values, except for the outliers (circles) and extremes (asterisks).

**Figure 2 animals-13-01014-f002:**
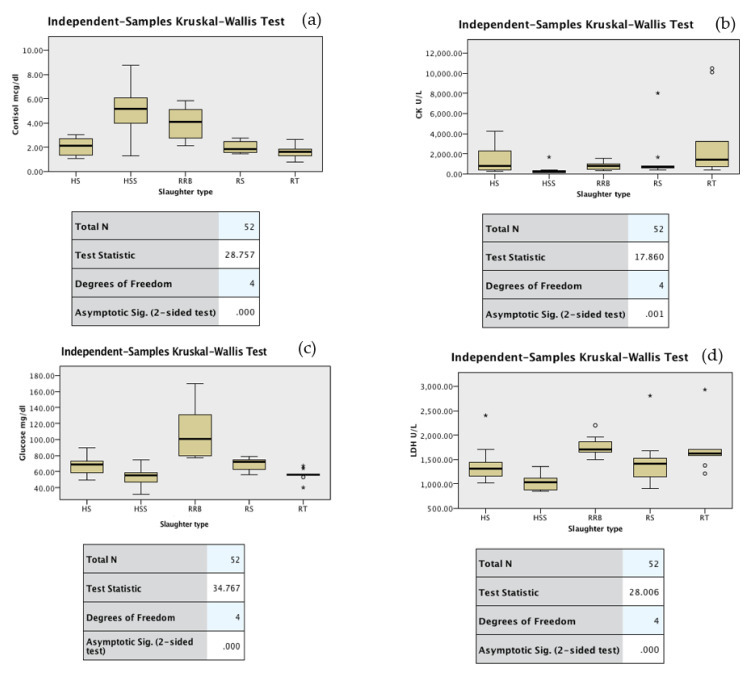
Box and whisker plots for cortisol (**a**), CK (**b**), glucose (**c**), and LDH (**d**) levels comparing halal slaughter (HS), halal stunning slaughter (HSS), rotating restraint box (RRB), regular slaughter (RS) and resting (RT). In the boxplot, the thicker line in the middle is the median value. The top and bottom box lines show the first and third quartiles. The whiskers show the maximum and minimum values, except for the outliers (circles) and extremes (asterisks).

**Figure 3 animals-13-01014-f003:**
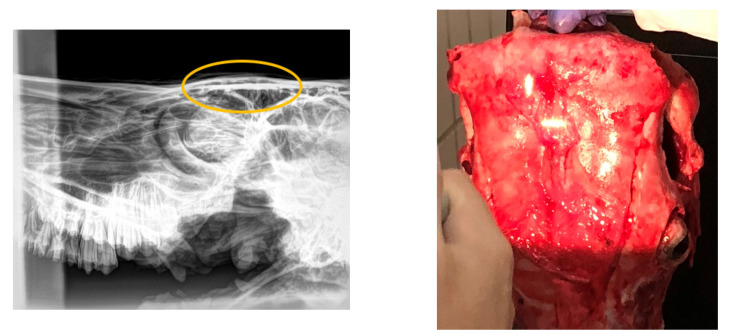
Digital radiography rating 1 (following Malaysia Standard 1500:2009), and related photograph.

**Figure 4 animals-13-01014-f004:**
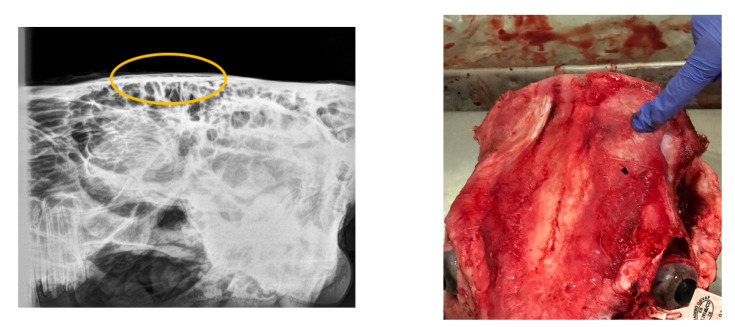
Digital radiography rating 2 (following Malaysia Standard 1500:2009), and related photograph.

**Figure 5 animals-13-01014-f005:**
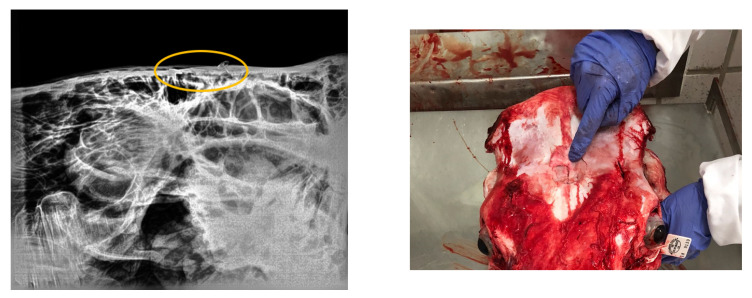
Digital radiography rating 3 (following Malaysia Standard 1500:2009). The eclipsed area defines the perimeter of the damaged area. On the right is shown the related photograph.

**Table 1 animals-13-01014-t001:** Data set for microbiological parameters.

Aerobic Mesophilic Bacteria (CFU/g)				Enterobacteria (CFU/g)			Coliforms (Microorg/g)		
	HS	RS	HSS	HS	RS	HSS	HS	RS	HSS
**Mean**	2753.3	593.3	223,525.1	10,960.5	19,137	388.5	6764.8	14,112.1	748.3
**SD**	5498.29	499.86	375,490.25	14,436.06	15,367.42	214.79	16,702	21,067.95	1260.55
**Max**	18,150	1355	1,100,000	50,000	40,000	800	53,000	58,860	4197
**Min**	80	170	185	1175	1500	100	30	200	30

Halal slaughter (HS); regular slaughter (RS); halal stunning slaughter (HSS).

**Table 2 animals-13-01014-t002:** Data set for physiological parameters.

	Glucose mg/dl					Cortisol mcg/dl					LDH U/L					CK U/L				
	HS	RS	HSS	RT	RRB	HS	RS	HSS	RT	RRB	HS	RS	HSS	RT	RRB	HS	RS	HSS	RT	RRB
**Mean**	68.1	69	53.4	55.8	109.5	2	2	5.1	1.6	4	1399.9	1463.5	1027.9	1701.4	1757.1	142.1	1455	378.1	3256.4	779.3
**SD**	11.15	8.31	13.5	7.1	33.49	0.7	0.51	2	0.5	1.27	404.96	527.48	153.6	459.4	192.7	1378.96	2334.2	464.3	3847.7	359.57
**Max**	90	79	75	67	170	3.01	2.8	8.8	2.63	5.86	2400	2804	1349	2931	2198	4232	8022	1672	10,482	1560
**Min**	49	56	32	55.82	77	1.08	1.46	1.3	0.8	2.15	1016	911	849	1211	1500	270	380	132.0	364	333

Halal slaughter (HS); regular slaughter (RS); halal stunning slaughter (HSS); resting (RT); rotating restraint box (RRB).

## Data Availability

Not applicable.

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
