# Peer review of "Influence of the Slaughter Method on the Hygienic Quality of Beef Cattle Meat and Animal Welfare Biomarkers"

_animals, 2023, doi:10.3390/ani13061014_

Round 1

Reviewer 1 Report

General comment:

The paper provides interesting information on the different methods of cattle slaughter, providing valuable information about them. The paper, with minor changes, can be accepted in my opinion.

Specific comment

Lines 29-30. Specify the kind of samples (plasma or blood?)

Lines 27 29. Clarify the total amount of samples. It seems that n=52 but it is not clear.

Lines 31-34. Reorder theses lines because conclusion have to close this paragraph. So, it is better insert “Moreover, it was carried out a radiological study of the possible skull damage due to the pneumatic gun used at the time of stunning in the halal rite slaughter” before that “ The halal rite slaughter using stunning with pneumatic gun resulted in the 31 best hygienic quality of meat and obtained the lowest amounts for all animal welfare biomarkers”.

Line 74. repeated research has demonstrated or several (o many) studies have demonstrated….and that there is no difference in animals, maybe between animals?

Line 157. Specify the signification level chosen.

Line 211. Delete  the point after sig ( sig.< 0.05).

Line 214. Change CPK, LDH and glucose contents by plasma CPK, LDH and glucose.

Line 215. Incorporated plasma before cortisol.

Line 216. Change blood by plasma ( if  you measure cortisol in plasma).

Line 217. Incorporate plasma before glucose results (probably you should delete the term results).

Line 352. It seems that something is lack at the beginning of the paragraph. Rewrite.

In theses lines I think that authors should point out the requirements, or complete the statements as it follows: “the compliance of any method of stunning depends on whether the animal remains alive (is able to recover and live a normal animal life if not slaughtered) following the stunning (Hayat Mustaqirrat) and prior to slaughter  or not, whether the act of stunning in itself is painful or not to the animal being stunned, and whether or not the stunning affects the flow of blood after slaughter— if it meets all these requirements, then it is permissible.”

Author Response

Dear reviewer,

Thank you very much for your comments. We have improved the manuscript following your valuable recommendations, which we list point by point in attached document. 

Thanks so much. 

Prof. Dr. Francisco Requena. 

Reviewer 2 Report

I have completed my evaluation of your manuscript. My evaluation is that you have an interesting study, which could justify for further evaluation and review. However, you must put considerable efforts into improving the manuscript. I have listed specific comments directly on the manuscript (attached). 

Author Response

Dear reviewer 2,

Thank you very much for your comments. We have improved the manuscript following your valuable recommendations, which we list point by point in the attached document. 

Kind Regards. 

Thank you. 

Reviewer 3 Report

Dear authors, please find my remarks to your manuscript bellow:

Why do the authors use ‚bovine’, when the study was done on beef cattle? Please consider changing throughout the text.

L 15: Please rephrase: Three slaughter procedures were compared, as follows: regular slaughter....

L21-24: Please rewrite the aim, since contains syntax errors (e.g. objectives of this research was to carry, types of slaughter) and also it is long and hard to follow.

L26-27: When we state that differences were found, we include the P value, to prove such statements; Same observation for L30-31.

L 27-28: Please rewrite this sentence, it makes little sense;

L 33-34: Is this a conclusion? It looks to me as a methods sentence, please remove it.

Abstract section: Please rewrite the entire section, and structure it as: aim, materials and methods, results and main conclusion.

Lines 94-95: Please include data on animals breed, where the animals of the same breed, or crossbreed, were animals matched in size and age?, all these factors.

Lines 120-123: Why were the 20 additional blood samples added? I would imagine that the different animals used and different handling conditions would influence the parameters studied. Plus, how is 12 samples and 10 samples leading to 20 samples added, and not 22 extra samples?

One of my main concerns is that the study does not present an ethics and animal welfare sub-section, especially since it is dealing with animals that are being slaughter, and claims to evaluate animal welfare. Was this study approved by the University of Cordoba? Were the animals handled with respect to EU Directive 2010/63 and/or Spanish national legislation? This aspect in my opinion is unacceptable for such a study.

L149-150: Please rephrase this sentence, there are syntax errors;

L 160: ‘The results revealed that statistical differences were observed’ please rephrase this to sound  ‘The slaughtering method was showed to influence (p0.05) all parameters, except for coliforms counting…’

Tables seem to me as print screens, please remediate this.

L 406-407: Please remove this  sentence, since it does not sound like a conclusion

Another concern for me is the low number of animals used (10/group), in order to support the conclusions and have soundness, I strongly believe that the sample size should have been larger.

Author Response

Dear reviewer,

Thank you very much for your comments. We have improved the manuscript following your valuable recommendations, which we list point by point in attached document. 

Kind regards. 

Thank you. 

Prof. Dr. Francisco Requena 

Round 2

Reviewer 2 Report

Dear author,

I have listed specific comments in the attached document. Caution is need to make comparisons, as you use different animals for each one of the outcome (pysiological, radrigraphy, meat quality).

Reviewer 3 Report

Dear authors, thank you for making all the changes that I suggested. Good luck with your work!

Author Response

Thank you so much for your comments.